# Enhanced Hippocampus-Nidopallium Caudolaterale Connectivity during Route Formation in Goal-Directed Spatial Learning of Pigeons

**DOI:** 10.3390/ani11072003

**Published:** 2021-07-05

**Authors:** Meng-Meng Li, Jian-Tao Fan, Shu-Guan Cheng, Li-Fang Yang, Long Yang, Liao-Feng Wang, Zhi-Gang Shang, Hong Wan

**Affiliations:** 1School of Electrical Engineering, Zhengzhou University, Zhengzhou 450001, China; mengmeng_li@gs.zzu.edu.cn (M.-M.L.); taotao_fan@163.com (J.-T.F.); CSG17839194958@163.com (S.-G.C.); flyer1014@163.com (L.-F.Y.); longyang_zzu@163.com (L.Y.); liaofeng_wang@163.com (L.-F.W.); 2Henan Key Laboratory of Brain Science and Brain-Computer Interface Technology, Zhengzhou 450001, China; 3Institute of Medical Engineering Technology and Data Mining, Zhengzhou University, Zhengzhou 450001, China

**Keywords:** hippocampus, nidopallium caudolaterale, route formation, goal-directed spatial learning, functional connectivity, pigeon

## Abstract

**Simple Summary:**

A distributed brain network supports the goal-directed spatial learning of animals, in which the formation of the route from the current location to the goal is one of the central problems. To enhance our understanding of how the avian hippocampus (Hp) and nidopallium caudolaterale (NCL) cooperate during route formation, we examined neural activity in the Hp-NCL network of pigeons and explored their connectivity dynamics in a goal-directed spatial task. We found that pigeons’ behavioral changes during route formation are accompanied by modifications of their neural patterns in the Hp-NCL network. The depressed spectral power in Hp and NCL, together with the different dynamics of the functional connectivity in both regions, as well as, most importantly, the enhanced Hp-NCL theta functional connectivity, provide insight into the potential mechanism of avian spatial learning.

**Abstract:**

Goal-directed spatial learning is crucial for the survival of animals, in which the formation of the route from the current location to the goal is one of the central problems. A distributed brain network comprising the hippocampus and prefrontal cortex has been shown to support such capacity, yet it is not fully understood how the most similar brain regions in birds, the hippocampus (Hp) and nidopallium caudolaterale (NCL), cooperate during route formation in goal-directed spatial learning. Hence, we examined neural activity in the Hp-NCL network of pigeons and explored the connectivity dynamics during route formation in a goal-directed spatial task. We found that behavioral changes in spatial learning during route formation are accompanied by modifications in neural patterns in the Hp-NCL network. Specifically, as pigeons learned to solve the task, the spectral power in both regions gradually decreased. Meanwhile, elevated hippocampal theta (5 to 12 Hz) connectivity and depressed connectivity in NCL were also observed. Lastly, the interregional functional connectivity was found to increase with learning, specifically in the theta frequency band during route formation. These results provide insight into the dynamics of the Hp-NCL network during spatial learning, serving to reveal the potential mechanism of avian spatial navigation.

## 1. Introduction

Spatial navigation is crucial for the survival of animals and is closely related to their migration, foraging, and homing [1]. As a typical form of navigation, goal-directed navigation has been widely studied. In goal-directed navigation, the three central problems that animals must solve are ‘Where am I?’, ‘Where is the goal?’, and ‘How can I get to the goal?’ (i.e., ‘What is the route from the current location to the known goal location?’) [2,3].

Existing evidence has shown that a series of specific neural substrates in the brain support animals in realizing this complex cognitive process. The hippocampus (Hp) is the most important target brain region in the study of spatial navigation mechanisms. A series of spatial function-related cells including the most famous place cell found in Hp [4,5,6] provide the internal physiological structure basis for the representation of the external environment. Specific neural oscillation activities contribute to the neural information suitable for encoding the current location and local trajectory on a fast time scale, which is very important for the decision-making behavior of animals in the spatial environment [2,7]. These neural representations help animals to form a ‘predictive map’ in the brain for route planning of navigation [8]. For the major area of high-level cognition, the prefrontal cortex (PFC), a large number of neurons in it represent behavior-related spatial information at the ensemble level [9]. Further study has shown that some prefrontal principal neurons encode the behavioral goal by increasing their discharge probability specifically during spatial navigation to guide the adaptive behavior of animals [10]. Consequently, these studies have indicated that, in mammalian spatial navigation, a local network composed of multiple brain regions including Hp and PFC plays an important role. These regions interact and cooperate with each other to share the processing and communication of spatial information across them through a temporal coordination mechanism modulated by specific neural oscillations [11], which supports the internal spatial representation of the external environment and flexible decision-making behavior.

Birds have excellent navigation capabilities and recent studies have indicated that avian canonical pallial circuitry is similar to its mammalian counterpart and might constitute the structural basis of similar neuronal computation [12], although the avian pallium seems to lack an organization akin to that of the mammalian cerebral cortex. This kind of structural similarity may provide convincing explanations for why many birds demonstrate sophisticated perceptual and cognitive behaviors [12,13]. Research has shown that the avian Hp and mammalian Hp are functionally similar [14], although they possess very different structures and the correspondence between their subdivisions is still debated [15,16]. In avian Hp, one set of spatial cognitive function-specific cells [17], including location cells [18], path cells [19], pattern cells [20], and so on, have been found. These cells constitute the neural basis for avian spatial cognitive function to learn the spatial locations and their relationships and make avian Hp the most important brain region for spatial navigation behavior [21]. Since 1982, a growing body of research has accumulated supporting the notion that the avian nidopallium caudolaterale (NCL) and mammalian PFC are analogous structures involved in performing high-order cognitive processes [22,23,24]. Researchers believe that the avian NCL may play a key role in navigation as a higher cognitive structure that sets goals, selects appropriate actions, and alters intermediate strategies when new and unexpected information becomes available [21]. Experimental results of pigeons performing a goal-directed task highlighted the decision-making function of NCL [25]. A further hypothesis has been proposed that avian Hp and NCL have a cooperative relationship in goal-directed decision-making tasks, in spite of the fact that it is inconclusive about the connection between them. We have reviewed related studies about the cooperative interaction in the avian Hp-NCL local network supporting the goal-directed routing information encoding and suggested its potential role in spatial learning [26]. Although most studies believed that there were rare direct links between avian NCL and Hp [27], a recent study that recorded the neural signals from both the Hp and NCL of pigeons performing goal-directed decision-making tasks demonstrated the existence of causal functional interactions between them [28]. During the process of spatial navigation, Hp and NCL provide various contributions and work together through a potential interaction mechanism to help birds to optimize the route to the goal progressively. To date, the neural mechanisms, especially the Hp-NCL connectivity patterns underlying the route formation in goal-directed spatial learning, are still unclear.

Changes in the internal states of the brain have been proposed to systematically influence how behavior improves with learning [29] and functional connectivity analysis between different brain regions is frequently used to measure the functional coupling of distributed neural systems [30] and multiple individuals [31]. We hypothesize that connectivity between the avian Hp and NCL might support the acquisition of route formation in goal-directed spatial learning, and we design the goal-directed spatial learning task of pigeons and analyze the behavioral and neural data in the task, hoping to explore the role of Hp-NCL connectivity dynamics during route formation. Here, we report that route formation during spatial learning in goal-directed behavior is associated with enhanced Hp-NCL functional connectivity, together with modifications in the connectivity of the Hp and NCL local functional networks, as well as the depressed spectral power in both regions. These results support the role of the Hp-NCL network during the acquisition of route formation in goal-directed spatial learning.

## 2. Materials and Methods

### 2.1. Subjects, Surgery, and Electrode Implantation

Pigeons (*Columba livia*) weighing 400 to 500 g were used in this study (*n* = 6). They were housed in an animal facility with a size of 3 m × 3 m × 2 m for at least 2 weeks before the experiments in the designed maze (Figure 1a), and it had plenty of sunlight, good ventilation, and free access to water and food. All experimental procedures related to animal experimentation were approved by the Life Science Ethical Review Committee of Zhengzhou University. Efforts were made to minimize the number of animals used and their suffering.

All surgeries were performed after the pigeons were anesthetized with 1.5% pelltobarbitalum natricum (0.25 mL/100 g) injected in intraperitoneal. The 16-channel recording microelectrode arrays (4 × 4 arrays, Hong Kong Plexon Inc., Hong Kong, China; Figure 2a) were chronically implanted at a location directly in the left Hp (AP 4.5 mm; ML 1.0 mm; DV 0.5 to 1.5 mm) and NCL (AP 5.5 mm; ML 7.5 mm; DV 2.0 to 3.0 mm) after the pigeons were placed in a stereotaxic apparatus, according to coordinates obtained from the Karten and Hodos stereotaxic atlas of the pigeon brain [32].

### 2.2. Goal-Directed Spatial Learning Experiment

After a recovery period of around one week, the pigeons implanted with electrode arrays were trained for a route formation spatial learning task in amaze. At the beginning of the experiment, pigeons were placed in the starting position (40 cm × 30 cm × 35 cm) as a waiting area; then, the animal was trained to explore the maze and find the goal with a food hamper through a route. The trial ended when the pigeon arrived at the goal and obtained the food reward or after 2 min elapsed. When the pigeon enjoyed the food reward at the goal, the food hamper at the goal was closed and the pigeon was allowed to return to the waiting area for the next trial, and this trial was recorded as a correct one. If the pigeon did not find the goal within 2 min, the experimenter guided the pigeon to the goal. For every pigeon, multiple spatial navigation trials per learning stage during four consecutive stages (S1, S2, S3, and S4) were applied, which were defined by different behavioral performance. Each session was composed of the above-mentioned four consecutive stages. There were two other positions besides the starting position with the food hamper in the maze, and one of them was set as the goal randomly for each session. In general, pigeons could gradually improve the route repetition rate through four continuous days of learning to form a stable route in each session. However, each pigeon showed differentiated learning performance; thus, for some sessions, in which the repetition rate of consecutive days was very close during learning, we included them in the same stage (except for S3 and S4). Finally, the whole learning process was integrated into four progressive stages in one session according to the performance. When the pigeon could reliably perform the task through a preferred path, in which the route repetition rate reached more than 80% of the total trial numbers for two consecutive days, it was considered that the experiment was completed.

### 2.3. Behavioral and Neural Data Recording and Analysis

Behavioral performance was analyzed by measuring the latency time to find the goal and route repetition rate, defined as the percentage of trials that the pigeon completed through the preferred route. The timings of the pigeons were obtained by the infrared detectors distributed on all of the pathlets along the routes in the maze. The animal trajectory was recorded by the observation camera placed on the ceiling and stored in the computer during the experiment.

A 128-channel Cerebus^TM^ Multichannel Acquisition Processor (Blackrock Microsystems, Salt Lake City, UT, USA) was used to record local field potential (LFP) signals from the Hp and NCL of the pigeon. The LFPs with a sampling rate of 2 kHz were filtered by a 0–250 Hz Butterworth low-pass filter. Power spectral density (PSD) and coherence were computed using multitaper Fourier analysis. Mean spectral power measures were calculated for different bands for the entire session of the task, including delta (1 to 4 Hz), theta (5 to 12 Hz), beta (13 to 30 Hz), slow-gamma (31 to 45 Hz), and fast-gamma (55 to 80 Hz). In this study, data from 16 sessions of six pigeons (numbered by P097, P098, P099, P100, P101, P103) were acquired and analyzed. In addition, some channels of the pigeons were detected as bad channels caused by detached electrode contacts, intermittent electrical connection, or line noise. These bad cannels were excluded from the analysis sequence.

### 2.4. Functional Network Connectivity Analysis

Coherence was used to measure the degree of synchronization [33] between the LFPs corresponding to the channels in Hp and NCL, which was calculated as follows:(1)$$Coh_{x,y}(f)=\frac{{\left|p_{x,y}(f)\right|}^{2}}{\left|p_{x}(f)\right|\times \left|p_{y}(f)\right|},$$
where

(2)$$p_{x,y}(f)=\frac{1}{n}\sum_{i=1}^{n}x_{i}(f)y_{i}^{*}(f),$$

For a given frequency f
, $p_{x}(f)$
and $p_{y}(f)$
represent the auto-power spectra of two LFP time series,x
and y
,respectively, and $p_{x,y}(f)$ is the cross-power spectrum.

We calculated the LFP coherence of each pair of channels from all channels in Hp, NCL, and between them. The coherence matrices of different frequency bands were obtained to construct the functional network. We binarized the above coherence matrices to visualize the network connectivity, setting different thresholds for the binarization according to experience. Note that, for the network between Hp and NCL, the inner connections in the single region (Hp or NCL) were removed and only the connections between them were retained.

The clustering coefficient, a topological characteristic of the functional network reflecting the intensity of the connection between different channels [34], was calculated, which refers to the possibility that the spatially adjacent channels of a certain channel in the network are also adjacent to each other. Taking a social network in life as an example, it can be compared to the possibility that one’s friends are also friends with each other. The functional adjacencies between channels are determined by the binarized network. It is often used in combination with other topological parameters to measure the small-world properties of networks and could be calculated for statistical analysis. The calculation formula is as follows:(3)$$clustering\;coefficient=\frac{1}{M}\sum_{i=1}^{M}\frac{2E_{i}}{k_{i}\left(k_{i}-1\right)},$$
where $k_{i}$
indicates that the i-th node has k edges connected with the other channels. $E_{i}$ indicates the number of connections in the network that connects with the i-th channel. M
is the total number of channels in the network. The value of the clustering coefficient ranges from 0 to 1. The larger the clustering coefficient, the higher the modularity of the network.

### 2.5. Statistical Analysis

Statistical analysis and result visualization were performed with SPSS Statistics (IBM, New York, NY, USA), Graphpad Prism (GraphPad Software, San Diego, CA, USA) and Matlab (The Mathworks Inc., Natick, MA, USA). Friedman analysis of variance (ANOVA)with multiple post-hoc pairwise comparisons was used based on the SPSS package automatically. Statistical results were presented as mean ± standard deviation (std), with the statistics including Chi-square value, degrees of freedom (df), and *p* value. Statistically significant results were indicated by the *p* value, and all of the *p* values in the post-hoc pairwise comparisons were adjusted by Bonferroni correction. Differences were considered significant for *p* < 0.05.

## 3. Results

### 3.1. Behavioral Performanceduring Route Formation

Pigeons were trained in the maze, a goal-directed spatial learning task in which pigeons learn the route to the goal with a food reward after several exploration trials (Figure 1a). In this study, 16 sessions from six pigeons were obtained finally, in which different goals were set or different routes were formed, and no session was identical to another. For the sessions of the same pigeon with the same goal, only the ones with different stable routes were retained for analysis. Animals correctly learned the task, as their route repetition rate progressively increased over time (*p* < 0.001, Chi-square = 59.918, df = 3; Friedman ANOVA; *n* = 6 animals, 16 sessions; detailed results of multiple post-hoc pairwise comparisons in Figure 1b). Similarly, the time spent by the pigeons per trial significantly decreased across stages (*p* < 0.001, Chi-square = 37.868, df = 3; Friedman ANOVA; *n* = 6 animals, 16 sessions; detailed results of multiple post-hoc pairwise comparisons in Figure 1c). Altogether, these data suggest that pigeons learned to solve the task, generating a preferred route to the goal.

### 3.2. Dynamics of Hippocampal Activity during Learning

Previous studies have shown that the avian Hp is functionally similar to that of mammals, and it plays an important role in path planning and adjustment of the spatial learning process. In this study, we explored the neural response patterns of the pigeons recorded from their Hp during a goal-directed spatial learning task (Figure 2).

We found differences in the power of hippocampal activity across learning stages (Figure 3a), and decreasing time-frequency energy ratio in the Hp was also taking place at almost all bands during the spatial learning (*n* = 6 animals, 16 sessions; Figure 3b). In addition to time-frequency characteristics, enhanced hippocampal theta (5 to 12 Hz) functional connectivity was detected (*p* < 0.05, Chi-square = 9.127, df = 3; Friedman ANOVA; *n* = 6 animals, 16 sessions; detailed results of multiple post-hoc pairwise comparisons in Figure 3c), consistent with decreasing energy occurring during the learning. We detected no progressive association between hippocampal connectivity and the learning stage in the other four bands (*p* > 0.05, Chi-square = 5.630 for delta, 4.843 for beta, 7.952 for slow-gamma, 5.012 for fast-gamma, df = 3; Friedman ANOVA; *n* = 6 animals, 16 sessions; Figure 3c). Hence, these results suggested that the spatial learning process modulated the dynamic network connectivity pattern in the hippocampal theta band.

### 3.3. Dynamic Neural Patterns in NCL during Learning

In the brain of the bird, NCL is compared to the avian “prefrontal cortex”, which plays an important role in adaptability, flexible behavior, and executive function. We recorded the neural activity of the pigeon from their NCL during spatial learning simultaneously (Figure 2). Similarly to Hp, we also found depressed spectral power in the NCL across learning stages (Figure 4a), and the time-frequency energy ratio progressively decreased over time with task acquisition (*n* = 6 animals, 16 sessions; Figure 4b). Next, we tested whether the spatial learning process was specific to the functional connectivity in some particular band. We identified changes in connectivity in all five frequency bands over time, and their clustering coefficients progressively decreasing over time with route formation (*p* < 0.001, Chi-square = 12.236 for delta, 77.288 for theta, 118.032 for beta, 92.204 for slow-gamma, 87.757 for fast-gamma, df = 3; Friedman ANOVA; *n* = 6 animals, 16 sessions; detailed results of multiple post-hoc pairwise comparisons in Figure 4c). These results revealed that the spatial learning correlated with the decreasing connectivity in NCL, but showed no specificity in any specific frequency band.

### 3.4. Evolution of Hp-NCL Connectivity during Learning

We considered whether LFP functional connectivity in the Hp–NCL circuit was modulated by task acquisition. We identified elevated interregional connectivity in theta (5 to 12 Hz) oscillations during goal-directed spatial learning (Figure 5a). We used the clustering coefficient characteristic of the functional network to measure the connectivity quantitatively. We found increasing connectivity in low bands (delta, theta, and beta) and decreasing connectivity in high bands (slow-gamma and fast-gamma). Importantly, we found prominent connectivity in the theta band (5 to 12 Hz; Figure 5b), which progressively increased significantly over time with task acquisition (*p* < 0.001, Chi-square = 22.467, df = 3; Friedman ANOVA; *n* = 6 animals, 16 sessions; detailed results of multiple post-hoc pairwise comparisons in Figure 5b). Also, similar increasing connectivity trends were also detected in delta band (*p* < 0.01, Chi-square = 13.739, df = 3; Friedman ANOVA; *n* = 6 animals, 16 sessions; detailed results of multiple post-hoc pairwise comparisons in Figure 5b) and beta band (*p* < 0.001, Chi-square = 22.224, df = 3; Friedman ANOVA; *n* = 6 animals, 16 sessions; detailed results of multiple post-hoc pairwise comparisons in Figure 5b); however, the other bands showed no significant changes (*p* > 0.05,Chi-square = 2.170 for slow-gamma, 6.927 for fast-gamma, df = 3; Friedman ANOVA; *n* = 6 animals, 16 sessions; Figure 5b) over time. Overall, these results suggested that spatial learning was associated with the progressive enhancement of Hp-NCL coupling in the theta frequency band specifically during route formation.

Increasing interregional connectivity may correlate with the changes in the amplitude of oscillatory activity. Therefore, we asked whether the Hp-NCL connectivity during spatial learning was modulated by the time–frequency characteristics of Hp or NCL. As discussed above, we found no significant difference specifically in the theta power of hippocampal activity (Figure 3a,b) or of the NCL (Figure 4a,b). All the detected associations between the spectral power and the learning stage in the other bands throughout acquisition showed decreasing trends (Figure 3a,b and Figure 4a,b), in contrast to the increasing trend detected in the Hp-NCL connectivity (Figure 5b). Hence, these results suggested that enhanced Hp-NCL connectivity during spatial learning was not dependent on the increase in spectral power; it was more likely the result of a shared pattern of change.

## 4. Discussion

In this study, we explored the behavioral performance of the pigeon and the neural patterns of recorded LFPs during a goal-directed spatial learning task. Our results show that route formation during spatial learning in goal-directed behavior was associated with enhanced hippocampus–nidopallium caudolaterale functional connectivity, as well as with increasing hippocampal connectivity, decreasing connectivity in NCL, and depressed spectral power in both two regions.

Our results demonstrated an accompanying strengthening of hippocampal functional connectivity in the theta band during route formation in spatial learning. Theta oscillation was first discovered in Hp and was proven to be inextricably linked to high-level cognitive processes, including memory encoding and working memory retention [35] in subsequent decades. For spatial-related information encoding, theta rhythms contribute a mechanism to coordinate related activities converging in Hp from multiple brain regions [36]. Benefiting from this kind of support for spatial memory encoding and retrieval, the hippocampal theta connectivity maintained a high level when the pigeons skillfully learned the path to the goal and formed the preferred route progressively. These results are consistent with previous research highlighting the important role of the theta band in spatial cognition. Different from the hippocampal results, the functional connectivity pattern of NCL did not show rhythm specificity during route formation but presented consistent suppression in all of the observed frequency bands. As a region involved in executive functions including decision-making and higher-order multimodal processing [37], NCL was suggested to work using different neural rhythms [38]. In our study, the pigeons’ burdens of real-time decision-making and multimodal information processing ability gradually decreased along with the deepening of familiarity with spatial tasks, which may have resulted in the neural activity depression in NCL.

There were limitations in the present study. First, our study was limited to the use of only the undirected analysis method to measure the hippocampus–nidopallium caudolaterale functional connectivity. We hypothesized that the interaction between Hp and NCL would support the acquisition of route formation in spatial learning. We observed that Hp-NCL connectivity in the theta band significantly increased selectively over time along with task acquisition, suggesting that it was correlated with spatial learning. Hippocampal–prefrontal synchrony has long been thought to be critical for spatial working memory [39], and previous results in rodents suggest that enhanced coupling of the Hp and PFC structures in the theta frequency range may constitute a general mechanism allowing them to interact selectively according to current behavioral demands [40]. Moreover, further findings indicated a theta-mediated mechanism of temporal coordination for shared processing and communication of spatial information across them, in which the theta-phase functioned to concurrently improve spatial representational accuracy in both regions [11]. For pigeons, Hp-NCL interactions have been demonstrated to exist in the goal-directed behavior [28], laying the foundations for future research on avian Hp-NCL interactions. We suggest that these similar results of functional interaction across species are bound to be closely related to structural similarity. On the other hand, a directional hippocampal–prefrontal interactions study reported that a strong, persistent lead from ventral Hp to medial PFC preceded an animal’s correct choice during a spatial working memory task [41], consistent with previous studies reporting the predominant direction of information flow from the Hp to PFC in awake behaving rodents [42]. Similar avian research also reported that the leading direction of information flow was from Hp to NCL when pigeons performed a goal-directed decision-making task [28]. Nonetheless, directional interaction analysis is still an area where much more work is necessary and further studies should be carried out. One idea is that cross-frequency coupling [43,44] should be introduced to explore the interaction between these two regions, providing another potential approach to understand their interaction mechanism under the premise that they may not support the same cognitive task using similar rhythms [38].

Secondly, this study did not examine and explain the relationship of the two different potential mechanisms for the spatial learning of the pigeons. There are two broadly opposing views regarding the mechanisms of animal spatial learning, namely response learning using a stimulus–response strategy and place learning using a cognitive strategy [45]. Current studies have suggested that spatial information is slowly incorporated into the optimal behavioral response in spatial learning [10], and response- and place-based frames may cooperate during route formation [46]. The internal topological representation of the external environment is involved in place learning relying on the Hp [47,48], while the task rules encompassing the association between the environmental stimulus and behavioral response are stored by response learning depending on the medial PFC [47]. The fact that both frames depending on distinct neural systems has solid substantial evidence in mammalian research, suggesting that the implementation of the route formation to the goal requires the interplay between distributed neural circuits. Specifically, Hp and medial PFC constitute the major brain regions popularly associated with place and response learning in this network. Similarly, the observed hippocampal representations, NCL correlations, and the interactions between them in this study are in line with the findings in mammals; thus, we could infer that the avian Hp and NCL enact the spatial learning task coordinately. Altogether, these results support the role of the avian Hp-NCL network during route formation in spatial learning.

Finally, we used the results from only six animals in this study. Although the limitation of sample size still existed, the 16 sessions from them alleviated this kind of limitation to a certain extent. In addition, a control group with a goal in random positions may make our results and conclusions stronger. Although the earlier stages of the experiment could be regarded as the control group to a certain extent, considering that the animals did not know that there was a goal in the initial learning, it may be more convincing to compare the results with those of the control group with more rigorous definition. This is a weakness of the current research, and we hope to solve this problem in the next step.

Overall, we have shown that behavioral changes in spatial learning during route formation are accompanied by modifications in functional connectivity in the Hp-NCL network, as well as changes in oscillatory power and a variety of functional connectivity in a single region. Altogether, our results provide insight into the dynamics of the Hp-NCL connectivity network during preferred route formation of spatial learning in a goal-directed spatial task. These changes in this local network are accompanied by the spatial learning process, serving to reveal the potential mechanism of spatial navigation.

## Figures and Tables

**Figure 1 animals-11-02003-f001:**
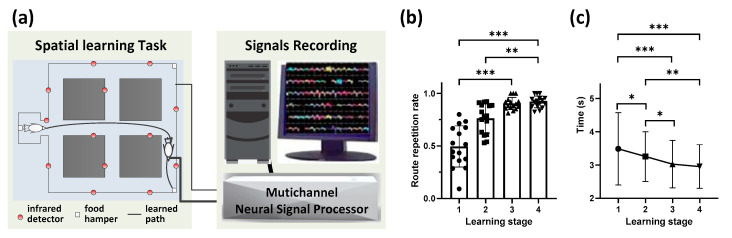
Behavioral performance during route formation in the maze. (**a**) Schematic diagram of the maze. Pigeons learn to find the goal with a food hamper full of grain food from the waiting area. After several navigation trials, pigeons learn the location of the goal. (**b**) Average route repetition rate per stage during route formation (** *p* < 0.01, *** *p* < 0.001; Friedman ANOVA; *n* = 6 animals, 16 sessions). Data are presented as mean ± std. (**c**) Average time spent per trial during route formation (* *p* < 0.05, ** *p* < 0.01, *** *p* < 0.001; Friedman ANOVA; *n* = 6 animals, 16 sessions). Data are presented as mean ± std.

**Figure 2 animals-11-02003-f002:**
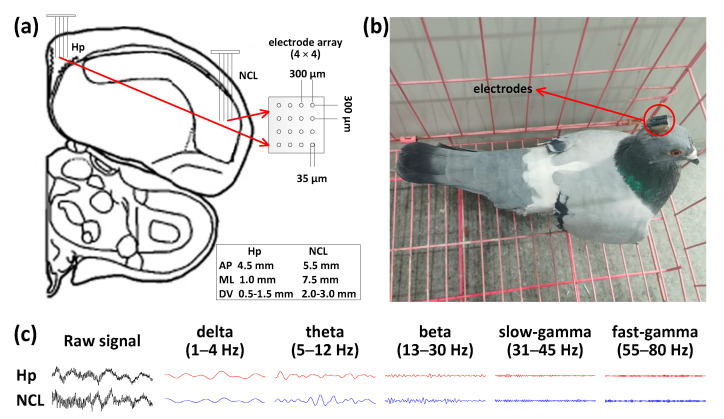
Implantation sites, microelectrode array, pigeon with implanted arrays, and LFP traces. (**a**) Diagram of implanting location Hp and NCL with the microelectrode array. AP: anteroposterior, ML: mediolateral, DV: dorsoventral. (**b**) Photograph of a pigeon (pigeon ID: P097) implanted with microelectrode arrays. (**c**) Examples of simultaneous LFP traces recorded from the Hp and NCL, filtered at delta, theta, beta, slow-gamma, and fast-gamma frequency bands (pigeon ID: P097).

**Figure 3 animals-11-02003-f003:**
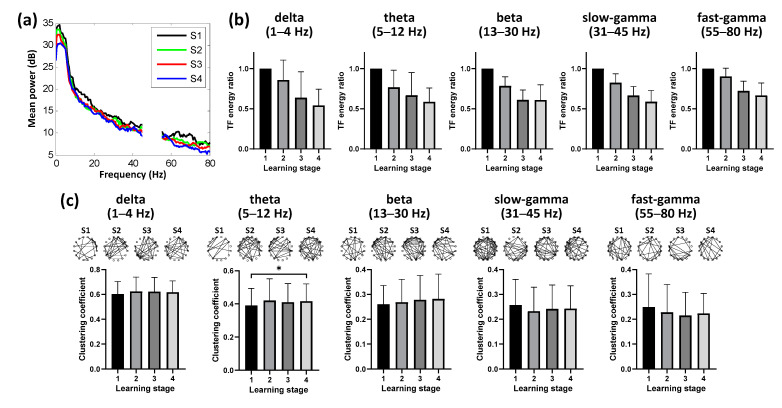
Dynamics of hippocampal activity during learning (*n* = 6 animals, 16 sessions). (**a**) Power spectral density in Hp during route formation of spatial learning. (**b**) Mean time–frequency (TF) energy changes in different bands relative to stage 1 for Hp. Data are presented as mean ± std. (**c**) Binarized hippocampal functional networks (Top, examples of the pigeon numbered P100) and statistical clustering coefficient results of the networks (Bottom, population data of all pigeons) for different bands across learning stages (* *p* < 0.05; Friedman ANOVA). Data are presented as mean ± std. S1 represents the learning stage 1, and so forth.

**Figure 4 animals-11-02003-f004:**
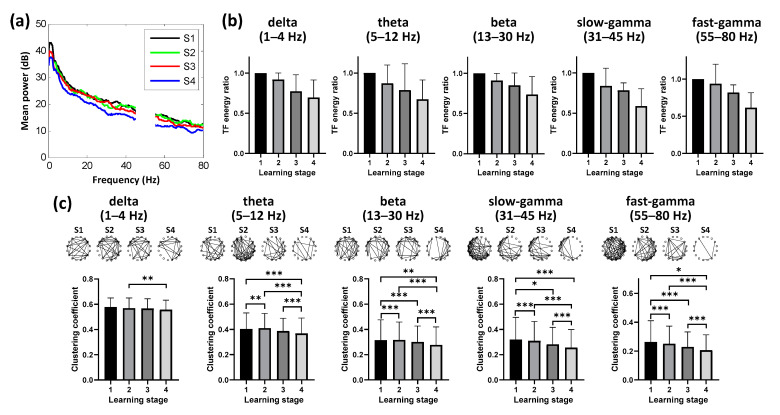
Dynamic neural patterns in NCL during learning (*n* = 6 animals, 16 sessions). (**a**) Power spectral density in NCL during route formation of spatial learning. (**b**) Mean time–frequency (TF) energy changes in different bands relative to stage 1 for NCL. Data are presented as mean ± std. (**c**) Binarized functional networks for NCL (Top, examples of the pigeon numbered P100) and statistical clustering coefficient results of the networks (Bottom, population data of all pigeons) for different bands across learning stages (* *p* < 0.05, ** *p* < 0.01, *** *p* < 0.001; Friedman ANOVA). Data are presented as mean ± std. S1 represents the learning stage 1, and so forth.

**Figure 5 animals-11-02003-f005:**
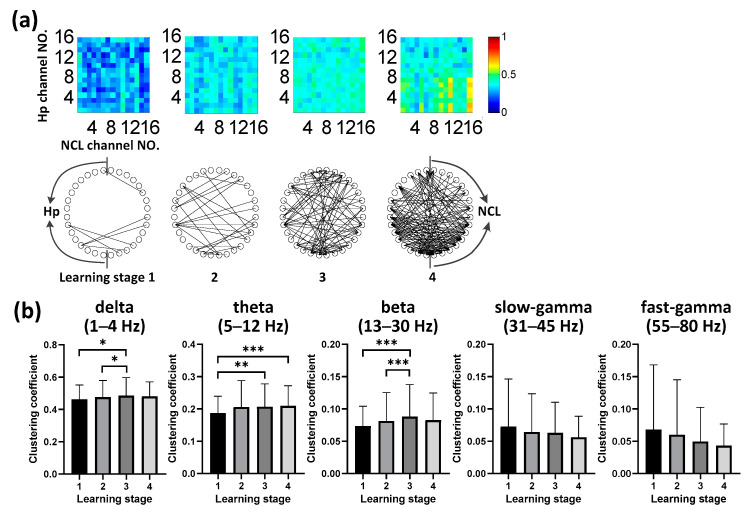
Synchronous activity between Hp and NCL during route formation in spatial learning. (**a**) Examples (from P100) of heatmaps of the Hp-NCL LFP coherence coefficient matrices and binarized functional networks for different learning stages in the theta band. (**b**) Clustering coefficient of functional Hp-NCL network (from all pigeons) for different bands across learning stages (* *p* < 0.05, ** *p* < 0.01, *** *p* < 0.001; Friedman ANOVA; *n* = 6 animals, 16 sessions). Data are presented as mean ± std.

## Data Availability

The data presented in this study are available on request from the corresponding author.

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
