# Peer review of "Enhanced Hippocampus-Nidopallium Caudolaterale Connectivity during Route Formation in Goal-Directed Spatial Learning of Pigeons"

_animals, 2021, doi:10.3390/ani11072003_

Round 1
Reviewer 1 Report
The manuscript by M-M. Li et al. is interesting because it presents neurophysiological data from awake behaving pigeons; such data are very rare in the avian neuro field, and the last author is clearly a leader in this subfield. The study is also interesting because it focuses on spatial learning and two highly studied avian brain regions, namely the hippocampus (Hp) and caudolateral nidopallium (NCL), which shares some similarity with mammalian prefrontal cortex. The data themselves are also useful (though not so easy to interpret). On the other hand, the manuscript does have some flaws. Although the English is very good for non-native speakers/writers, there are quite a few sections where it is difficult to understand. I was tempted to help correct these issues, but the lack of line numbers in the manuscript would make this very difficult for me. Therefore, I will only comment on the most severe language issues and, instead, focus my critique on the science. There, my principal complaint is that the methodology (experimental and analytical) needs to be described and discussed in more detail. Below, I first list my major concerns, followed by a list of less significant (mostly language-related) issues.
Major Issues:
1) The authors should be more careful in how they compare the avian Hp and NCL to mammalian brain regions. The avian Hp is clearly homologous to mammalian hippocampus (though some of the subdivisions are not so easy to homologize) but calling them “equivalent” goes well beyond the available data. I’m sure that I don’t want an avian Hp to replace my own hippocampus, for example. Even the term “functionally homologous” is also inappropriate, since biological homology often involves changes in function and we really have no idea whether Hp and its mammalian homolog share ALL of their functions (as far as I know, only its spatial memory function has been studied in birds). All of these concerns apply also to NCL, except that in this case most avian neurobiologists (nowadays) agree that NCL is NOT homologous to the mammalian prefrontal cortex. The authors are clearly aware of this, but the point is obscured when they call NCL “equivalent” to prefrontal cortex. I recommend simply to write that the avian Hp and NCL are structurally and/or functionally “similar” to mammalian hippocampus and prefrontal cortex, respectively.
2) It is unclear to me how the authors define/identify the 4 stages of learning; they describe the learning criterion for terminating the experiment, but not how the stages are identified. Is the number of trials needed to reach criterion divided into 4 equal sets? Maybe, but I can’t be sure. Alternatively, are the different stages defined by different performance criteria? I think it is important to be clear on this.
3) The procedure for “binarizing” the connectivity matrices is not described. At first I thought this procedure must involve some sort of thresholding, making me wonder how the thresholds were selected. However, I then read that the authors cite reference #34, which uses a measure called E-glob. Is this what the authors did here as well? The details should be specified and described in such a way that readers don’t have to read another paper to understand what was done. Along the same lines, I did not fully understand what the binarized network diagrams in Figs 3c, 4c, and 5a represent. Do the illustrated diagrams represent “examples” or population data. The legend for Figure 5 suggests that the former, but this is not clear for the earlier figures. If they are examples, how were they selected? Are they “best cases” for illustrating the population data from the clustering coefficient analyses?
4) The description of the clustering coefficient was not clear to me. What do the authors mean when they write that “it reflects the intensity of the connection between different channels”? Surely, there most be a better way to describe what this coefficient quantifies. The density of interconnections after binarization? I realize that more sophisticated readers will simply look at the formula and understand, but having a clear “plain language” description would be helpful (at least to me).
5) I would like to know more about the electrodes. The descriptor “4x4 array” wasn’t enough for me to figure out what the array actually looked like, and the company website didn’t help. I think it would be helpful, for example, to know the spacing and configuration of the contacts. I also wonder whether inserting the array might have cut any axons that, for example, interconnect different parts of the hippocampus or NCL.
6) I am concerned about the frequently repeated statement that the statistical analysis is based on 6 animals, 16 sessions, and 1,002 trials. The authors mention in the discussion that n=6 is on the low end (I agree) but that this problem is mitigated by analyzing 16 separate recording session. However, this raises the thorny issue of pseudoreplication: were the 16 session treated as independent data points when, in reality, they came from only 6 different birds and, therefore, were not really independent? The problem would be even worse if all 1,002 trials were treated as independent samples. At a minimum, I think the authors should state very clearly which N applies to each of their analyses; simply listing the same set of numbers (6, 16, 1002) for each analysis is unsatisfactory. Ideally, the data from each bird would be combined so that N=6 for each analysis.
7) The authors state that “all of the p values in the post-hoc pairwise comparisons were adjusted by Bonferroni correction,” which is commendable. However, the authors analyzed their data by learning stage and frequency band (delta, theta, beta, etc.), which means that they probably performed far more “multiple comparisons” than they accounted for with their Bonferroni correction. This becomes especially important for the analyses of clustering coefficients, where the bar graphs suggest that the effect sizes are really, really small (e.g., the “significant” difference between stages 1 and 4 in the theta band – shown in Fig. 3c – doesn’t really “look” significant to me; I would make the same argument for many of the supposedly significant differences illustrated in Fig. 4c). In other words, I am not convinced by many of the reported differences and suspect that they are statistically significant only because the authors did not fully account for multiple comparisons. I may be wrong about this, but I’d like the authors to convince me by presenting their statistical procedures in more detail.
8) In manuscript’s penultimate paragraph, the authors write that “Hp and PFC execute these two frames respectively,” with the two “frames” being place learning and S-R learning. However, I doubt that one can really say that PFC is not involved in allocentric place learning, given how closely it interacts with the hippocampus. I appreciate that the authors provide a reference for their claim, but that study refers specifically to lesions of the medial prefrontal cortex in rodents, rather than PFC generally. Therefore, I’d prefer a more cautious, nuanced statement.
9) I was confused by the labeling of the y-axis in Fig 1c as “Time/s” because I read that as “time per second”. Please write “Time (s)” instead. Also, the legend for this figure states that this represents “time spent per stage” but I strongly suspect that the authors mean “time spent per trial”. Please clarify. Similarly, in Figure 3a, I was confused by the “mean power/dB” axis label.
10) On page 7, the authors write that in NCL spectral power decreased across learning stages, except in the theta and beta bands. Statistically speaking, however, the authors have not demonstrate a lack of change in the theta and beta bands; they have simply shown that there is no statistically significant decrease, which is not the same thing. Indeed, looking at the graphs, power does seem to decrease across stages in those bands; it’s just that the changes don’t appear to be statistically significant.
Minor Issues:
1) It is redundant to write “progressively learned”; the authors can omit the “progressively”
2) In the sentence “Further study has shown that the prefrontal principal neurons encode the behavioral goal”, please replace “the prefrontal” with “some prefrontal”
3) The authors write “recent studies have indicated that avian canonical pallial circuitry .... might constitute the structural basis of neuronal computation.” I have no idea what is meant by the last part of this sentence. All neurons compute (something).
4) On page 8, replace “we found increased connectivity” with “we found increasing connectivity”. Similarly, replace “decreased” with “decreasing”. In the last sentence on this page, I recommend writing: “...not dependent on the increase in spectral power; it was more likely the result of a shared pattern of change”
5) The last paragraph on page 9 needs additional editing for English; it still has a lot of problems.
Author Response
Dear Reviewer,
Thank you for the thoughtful, helpful, and most kind reviews of manuscript animals-1271541 (Enhanced hippocampus-nidopallium caudolaterale connectiv-ity during route formation in goal-directed spatial learning of pigeon). The comments and suggestions have been incorporated as appropriate into the revised draft. The revised parts are highlighted in the revised manuscript. Specific revisions are shown in the attached file.
Yours sincerely,
Mengmeng Li

Reviewer 2 Report
The article is interesting, and introduction explains the theoretical background clearly and straightforwardly. The authors analyzed Hp-NCL connectivity in pigeons along trials in a spatial task. The authors found differences between sessions in behavioural performance and electrophysiological recording. Results are clear and the discussion remains appropriated. However, authors should include control groups to make stronger their results. For instance, they are not able to conclude electrophysiological recording is due to spatial learning. Before that, they should compare the electrophysiological recording in their animals with a control group, for example, a group with a goal in a random position. It is not suitable to conclude that electrophysiological results obtained in the manuscript are due to goal directed behaviour. I am not saying the conclusions were not true, I am trying to say that author should exclude other possibilities including a control group or a cue group before publishing it.
Author Response

(The authors gave the same response as above.)

Round 2
Reviewer 1 Report
I appreciate the authors’ attention to my comments. I only have a couple of issues that deserve more attention. Because the publisher has given me only 3 days for this re-review, I cannot be as thorough or helpful as I would like, but that is something the publisher needs to think about. In any case, here are my remaining suggestions:
1) line 29-30: I would write “it is not fully understood how the most similar brain regions in birds, the hippocampus....”
2) line 74: I would write “structural basis of similar neuronal computation”
3) line 144-45: I still wish the authors had been a bit clearer on how the four stages were defined. Figures 1b&c indicate that there is no significant performance difference between stages 3 and 4. So, how can those stages be discriminated?
4) line 146: replace “stating” with “starting”
5) line 186: I’d write “network connectivity, setting different thresholds for the binarization according to experience”
6) lines 191-193 and line 198: now I’m even more confused about how to understand the clustering coefficient. It represents the possibility that neighboring channels are neighbors? I suspect you mean something like “spatially adjacent channels are also adjacent in the matrices” but then you need to explain how you processed the matrices to get “functional adjacency” or something like that. Maybe I’m just too dumb to understand [I did manage to understand what clustering means in discussions of small-world networks, but that was a few years ago; is the meaning here analogous? line 199 suggests that it is, but what does “collectivization degree” mean?]. In any case, I suggest the authors ask someone who’s not already familiar with their clustering coefficient to help develop a clear and simple description of what it is/means.
7) line 280: I would write “ energy ratio showed no statistically significant...”
8) line 334: replace “form” with “from”
9) line 368: Here, too, you can safely delete “progressively”
10) line 380: I would replace “analogy” with “similarity” because some evolutionary biologists will understand “analogy” to refer only to functional similarity.
Author Response
Dear Reviewer,
Thank you for the quick, helpful, and most kind reviews of manuscript animals-1271541 (Enhanced hippocampus-nidopallium caudolaterale connectiv-ity during route formation in goal-directed spatial learning of pigeon). The comments and suggestions have been incorporated as appropriate into the revised draft. The revised parts are highlighted in the revised manuscript. Specific revisions are noted in the attached file.

Reviewer 2 Report
The ms has been improved. I would recommend to include in discussion the reasons why they did not included a control group in the experiment. I have no more concerns.
Author Response

(The authors gave the same response as above.)
